

# Opinion: Challenges and needs of tropospheric chemical mechanism development

Barbara Ervens[1,*], Andrew Rickard[2,3,*], Bernard Aumont[4], William P. L. Carter[5], Max McGillen[6], Abdelwahid Mellouki[6,7], John Orlando[8], Bénédicte Picquet-Varrault[4], Paul Seakins[9], William R. Stockwell[10], Luc Vereecken[11], and Timothy J. Wallington[12]

[1]Institute of Chemistry, University Clermont Auvergne, CNRS, 63000 Clermont-Ferrand, France
[2]Wolfson Atmospheric Chemistry Laboratories, Department of Chemistry, University of York, Heslington, York, YO10 5DD, 5 UK
[3]National Centre for Atmospheric Science, University of York, Heslington, York, YO10 5DD, 5 UK
[4]Univ Paris Est Créteil and Université Paris Cité, CNRS, LISA, 94010 Créteil, France
[5]College of Engineering Center for Environmental Research and Technology (CE-CERT) University of California, Riverside, CA 92521, USA
[6]Institut de Combustion, Aérothermique, Réactivité Environnement (ICARE), CNRS, 1C Avenue de la Recherche Scientifique, CEDEX 2, 45071 Orléans, France
[7]University Mohammed VI Polytechnic (UM6P), Lot 660, Hay Moulay Rachid Ben Guerir, 43150, Morocco
[8]Atmospheric Chemistry Observations and Modeling Lab, National Center for Atmospheric Research, P.O. Box 3000, Boulder, CO 80307, USA
[9]School of Chemistry, University of Leeds, Leeds, LS2 9JT, United Kingdom
[10]Department of Physics, University of Texas at El Paso, El Paso, Texas, USA
[11]Institute for Energy and Climate Research IEK-8: Troposphere, Forschungszentrum Jülich GmbH, 52425 Jülich, Germany
[12]Center for Sustainable Systems, School for Environment and Sustainability, University of Michigan, Ann Arbor MI 48109, USA
[*]These authors contributed equally to this work.

**Correspondence:** Barbara Ervens (barbara.ervens@uca.fr), Andrew Rickard (andrew.rickard@york.ac.uk)

**Abstract.** Chemical mechanisms form the core of atmospheric models to describe degradation pathways of pollutants and ultimately inform air quality and climate policy makers and other stakeholders. The accuracy of chemical mechanisms relies on the quality of their input data, which originate from experimental (laboratory, field, chamber) and theoretical (quantum chemistry, theoretical kinetics, machine learning) studies. The development of robust mechanisms requires rigorous and transparent procedures for data collection, mechanism construction and evaluation, and creation of reduced or operationally defined mechanisms. Developments in analytical techniques have led to a large number of identified chemical species in the atmospheric multiphase system which have proved invaluable for our understanding of atmospheric chemistry. At the same time, advances in software and machine learning tools have enabled automated mechanism generation. We discuss strategies for mechanism development, applying empirical or mechanistic approaches. We show the general workflows, how either approach can lead to robust mechanisms and that the two approaches complement each other to result in reliable predictions. Current challenges are discussed related to global change, including shifts in emission scenarios that result in new chemical regimes (e.g. low NO scenarios, wildfires, mega/gigacities) and require the development of new or expanded gas- and aqueous-phase mechanisms.



In addition, new mechanisms should be developed to also target oxidation capacity, and aerosol chemistry impacting climate, human and ecosystem health.

## 1 Introduction

The troposphere is a highly complex, dynamic chemical system composed of multiple phases (gaseous, aqueous, organic matrices) containing millions of chemical compounds that constantly mix and interact. The oxidation chemistry leading to the photochemical degradation of these compounds, resulting in the formation of a wide range of harmful secondary pollutants, including ozone and secondary organic aerosol (SOA) is complex and nonlinear. To comprehensively describe all chemical interactions, numerical models are indispensable, with chemical mechanisms that include kinetic and mechanistic information describing the formation, cycling and degradation of chemical compounds. Thus, chemical mechanisms merge information from a variety of fundamental research activities including laboratory, theoretical and field studies. The reactions that make up these mechanisms are incorporated in models in the form of ordinary differential equations allowing for the prediction of temporal changes in species concentrations due to chemical processing, which can be computationally challenging. Chemical mechanisms need to be applicable over a wide range of spatial and temporal scales (local, regional and global) and conditions, including indoor and outdoor environments. Chemistry models can be either applied separately (e.g., 'box models') or form modules in larger model frameworks that also consider additional processes, possibly in multiple dimensions, such as transport, dynamics, emissions and deposition, and convection.

The development of gas-phase chemical mechanisms describing the oxidation of volatile organic compounds (VOCs) can be traced back to the 1950s arising from interest in photochemical reactions resulting in high ozone levels in Los Angeles (Haagen-Smit, 1952). In the 1970s, acid rain was recognized as a major threat to ecosystems (Likens and Bormann, 1974), where the acidification was ascribed to the production of sulfate, nitrate and organic acids (Calvert and Stockwell, 1983). Research showed that gas-phase oxidation reactions alone were not sufficient to explain the measured acidification levels in rain and fog water, suggesting that aqueous phase conversion processes could not be ignored (Hoffmann and Jacob, 1984). These conclusions initiated the expansion of atmospheric mechanism development to include chemical processes in cloud droplets. The resulting coupled gas/aqueous ('multiphase') mechanisms in regional atmospheric models led to a comprehensive understanding of the relevant factors of precipitation acidity, and ultimately to policy measures on its mitigation (Grennfelt et al., 2020). The role of multiphase chemistry for chemical budgets in the atmosphere has since then been further explored and expanded towards the inclusion of oxidants and organic compounds (Abbatt and Ravishankara, 2023).

Recognition that a major fraction of atmospheric aerosol mass is secondary, i.e., formed by in situ chemical processes, combined with evidence of the effects of aerosols on climate, ecosystems and human health has stimulated research during the last decades (Pye et al., 2023). As a consequence, atmospheric chemical models have become increasingly more complex, in particular in terms of the consideration of multiple phases. At the same time, huge advances in theoretical, analytical and numerical techniques provided a wealth of detailed information on the variety, abundance and properties of organic species. The simultaneous development in computational power and capabilities allowed the implementation of increasingly complex



chemical mechanisms into models up to a point where it may become unfeasible to include all available information. Therefore, targeted and sophisticated strategies are needed to collect, filter, evaluate and use the relevant information to answer urgent and emerging research questions on the chemical composition of the atmosphere and its evolution in a changing climate and society. Such strategies will be only successful and efficient if reliable data are available and accessible, and their use and
implementation in chemical mechanisms is transparent.

In the current opinion article, we discuss the different strategies as used for chemical mechanism development and the individual steps and activities they comprise. We do not aim to provide a detailed review of the investigation or description of individual chemical systems; we refer the reader to the numerous (review) articles on these topics cited throughout the text. Instead, we frame our discussion based on the complementary, differences and applicability of the main mechanism
development strategies (Section 2). The spectrum of research activities in terms of theoretical and experimental studies that are required to inform mechanism developers of reliable fundamental data is described in Section 3, briefly summarizing recent advancements, current status and perspectives. In Section 4, we systematically outline the various steps needed to develop chemical mechanisms ensuring their reliability and robustness, together with an overview of progress and status for each individual step. We discuss insufficiently characterized chemical systems and regimes, their related challenges and the
emerging needs for future chemical mechanism development in Section 5. We conclude with an outlook on future needs and directions of chemical mechanism developments (Section 6).

## 2    Strategies of chemical mechanism development

The choice of the approach for mechanism development depends on (1) the intended purpose of the mechanism and (2) how well we understand the chemical system being represented. Figure 1 shows the approach if the mechanism is to be
used primarily to compile what we know about the system from a scientific perspective. This has primarily been used in the initial development of explicit 'master mechanisms' such as the Master Chemical Mechanism (MCM, https://mcm.york.ac.uk/MCM/), which has proven valuable in summarizing current knowledge, and for the analysis of detailed chemical systems, and mechanism development research. 'Master mechanisms' have been initially used to predict specific ozone relevant metrics such as maximum incremental reactivity (MIR), or photochemical ozone creation potentials (POCP), where they are particularly
useful for simulating multi-day photochemical ozone formation (Derwent and Jenkin, 1991; Carter, 1994).

Explicit mechanisms can be derived manually, as is the case for the NCAR Master Mechanism (Madronich and Calvert, 1989), MCM (Jenkin et al., 1997, 2003, 2015), CAPRAM, e.g. (Ervens et al., 2003; Hoffmann et al., 2020) and CLEPS (Mouchel-Vallon et al., 2017), or automatically as is the case for GeckoA, e.g. (Lannuque et al., 2018) and SAPRC-Mechgen (Carter, 2024a, b). Both methods are based on existing fundamental chemical data and theories, and rely on expert judgments
and estimates of mechanism developers. The difference is that automated methods require developing pre-defined estimation protocols for all reactions that may occur, while manual methods allow for separate examination/evaluation of the specifics of each chemical system. Regardless of how they are derived, these explicit mechanisms tend to be very large owing to the numbers of possible reactions, in particular for organic compounds. Manually derived mechanisms necessarily have to either





ignore or approximate relatively minor processes, which may not be negligible when taken as an aggregate. Even computer-
generated mechanisms require use of a "minimum yield" parameter to eliminate processes considered to be negligible and
also must use approximate methods, e.g., to represent cross-reactions of the many peroxy radicals predicted to be formed.
Therefore, most scientifically based mechanisms are necessarily 'semi-explicit' and reduced, at least to some extent.

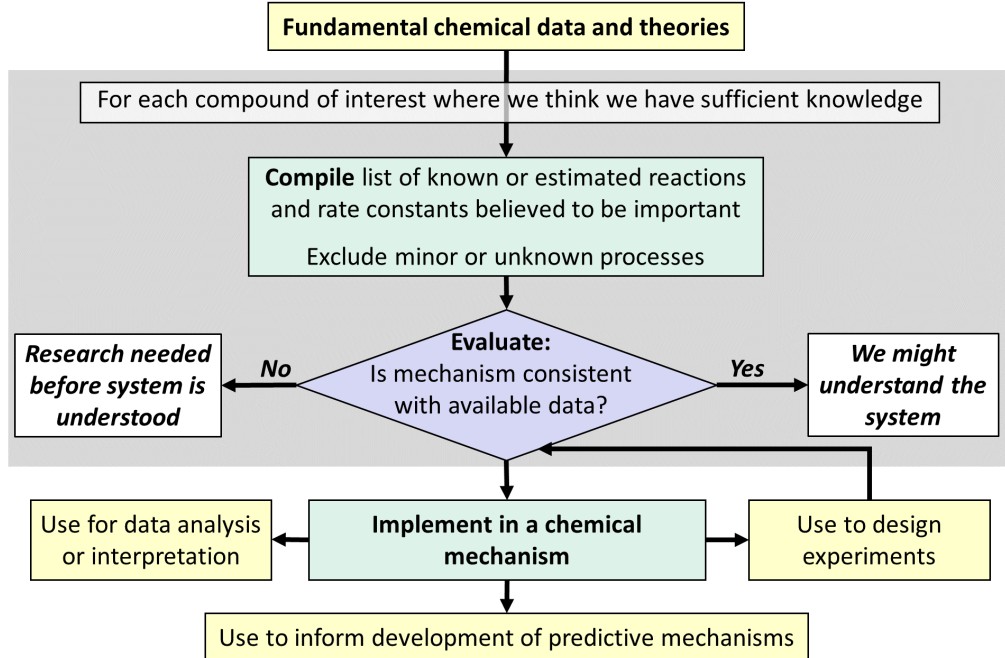

**Figure 1.** Outline of approach for development of (semi)explicit mechanisms to be used primarily to compile scientific knowledge or interpret
scientific data.

There is a valid concern that neglecting or over-simplifying chemical details in a model may introduce errors or biases
in its predictions. However, Stockwell et al. (2020) expressed the opposite concern that the creation of very large, explicit
mechanisms may involve an extensive over-extrapolation of the underlying structure-activity relationships (SARs). This may
occur when SARs are used to estimate unknown reaction parameters outside of an acceptable range of the chemical databases
used to derive them. These mechanistic uncertainties may ultimately propagate into the predictions based on the atmospheric
models. In fact, Puy et al. (2024) showed that increasing model detail greatly beyond current knowledge does not improve
model estimates and in the worst case make a model unverifiable. Therefore, a reasonable limit to the level of detail in a
chemical mechanism due to deficiencies in the available empirical and theoretical chemical databases should be acknowledged,
to appropriately inform public policy makers and other stakeholders. However, overly detailed mechanisms may provide strong
guidance for prioritizing experiments and quantum chemistry calculations for the further development of databases.

Explicit mechanisms, derived as indicated in Figure 1, generally do not attempt to represent chemical systems where we lack
sufficient understanding to justify estimating explicit reactions. Instead, chemical systems with large uncertainties are judged





to be areas where more research and evaluation are needed before they can be represented in scientifically based chemical mechanisms. Although valuable for scientific research, this is not sufficient for predictive modeling, which is necessary for assessing how air quality is affected by emissions, or what pollutants might be formed and how they evolve.

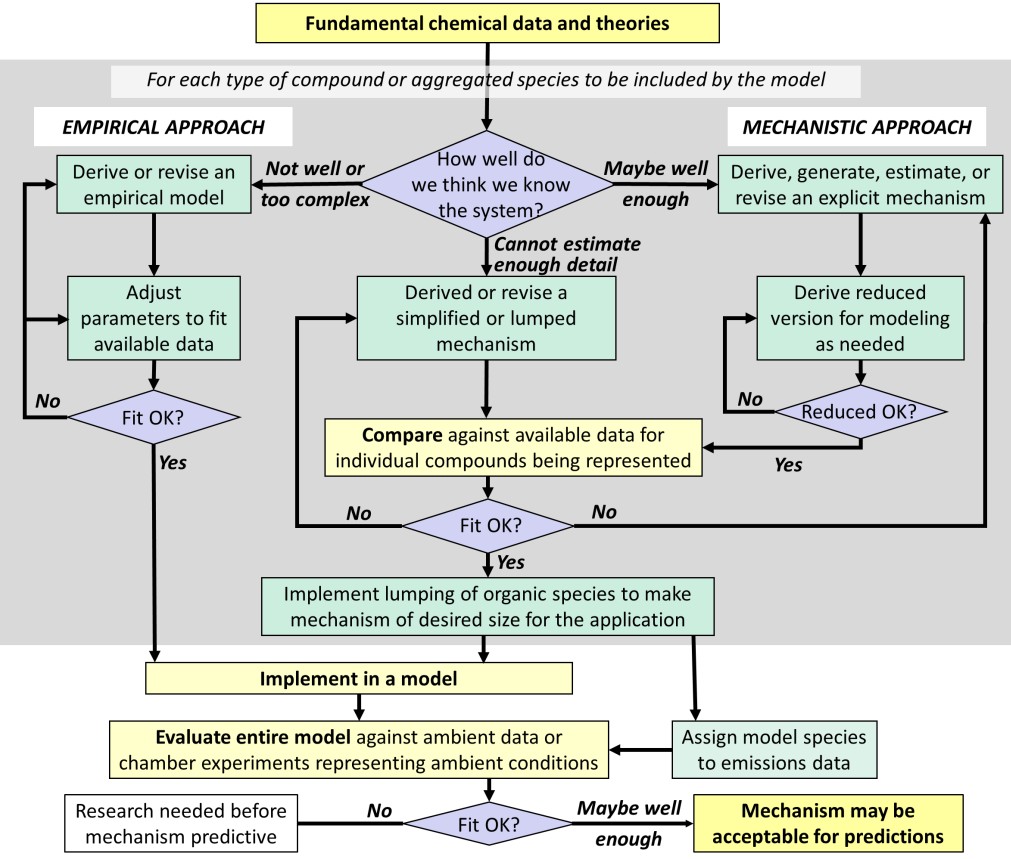

**Figure 2.** Outline of approaches for development of predictive mechanisms to be used for analyses of environmental impacts.

The need for a predictive modelling capability has been a major justification for continuing funding of mechanism development research. An outline of an approach for developing predictive mechanisms is shown on Figure 2. Ideally, we understand
the system sufficiently to be able to use the approach shown on the right side of Figure 2 for all portions of the mechanism, as advocated by Kaduwela et al. (2015). This would result in the most scientifically justifiable mechanisms that are least likely to have compensating errors, even if some uncertain rate coefficients have to be modified to improve fits to available data. However, we are not able to derive predictive explicit mechanisms for many types of compounds, so representing them requires use of (often highly) simplified or parameterized mechanisms with uncertain parameters adjusted to fit environmental chamber
data. In addition, the chemistry and physics of SOA formation and processing in both the gas and aqueous phases is sufficiently uncertain and complex that the use of largely parameterized models is necessary for any hope of predictive capability of spe-



cific SOA properties. In these cases, direct comparisons with experimental data and making adjustments to improve fits are a necessary part of mechanism development, but such adjustments are subject to uncertainties due to possible compensating errors, or due to the conditions of the experiment not representing the full range of conditions in the environment.

An additional consideration is that many more types of compounds are emitted into or formed within the atmosphere than can be represented explicitly in practical modeling applications, which makes some degree of aggregation or lumping necessary. This requires (1) developing mechanisms for lumped model species to represent a range of chemically similar compounds, and (2) developing rules for assigning these model species to specific compounds. The mechanisms for lumped model species could be based on those derived for selected representative compounds, or they can be constructed manually based on general

considerations. The former approach has the advantage that mechanisms for individual compounds can be evaluated using experimental data, since experiments cannot be carried out with lumped model species. In any case, the optimum set of lumped model species, and the rules for assigning compounds to them, will depend on the model application. Lumping approaches for most current mechanisms used in predictive models are generally based on considerations of $O_3$ predictions, which are not optimal for predicting SOA and other secondary pollutants.

Both strategies generally imply a sequence of similar steps to build high-quality chemical mechanisms that can be applied to a range of chemical systems and research questions (Section 3). The underlying research questions determine the focus and its potential importance. For example, a high molecular weight VOC present in low concentration may have a limited effect in controlling OH reactivity and secondary ozone formation, but might be key in new particle formation.

     These strategies as highlighted in Figures 1 and 2 are not exclusive but rather complementary. They have some fundamental

principle elements in common, namely the (i) need for fundamental chemical databases, (ii) mechanism generation, which may include the development and use of SARs and/or parameterizations, (iii) reduction of the mechanism to a manageable size and/or a specific focus and (iv) evaluation to ensure robustness and accuracy (Figure 3). Assessments of predicted differences from models using mechanisms based on the two approaches in Figure 1 may eventually lead to the identification of uncertainty and knowledge gaps that warrant further research. However, this requires that each individual step (or even intermediate

steps) is carefully performed taking advantage of and combining insights from various research activities (Section 4). Thus, a delicate balance between completeness and practicability should be the target so that models maintain their utility for users and stakeholders, including policymakers, while enabling easy and transparent incorporation of the most up-to-date information on the underlying chemical processes.

     Developments during the last decade in terms of targeted cheminformatics methods based on data mining, machine learning,

data assimilation and/or similar mathematical approaches may provide new opportunities for developing reduced mechanisms or optimizing lumping approaches. These will ultimately and efficiently lead to reliable modeling tools integrating the data from a range of concerted research activities.



# 3   Steps of chemical mechanism development

In general, mechanism development should follow five iterative steps, as outlined in Figure 3 and the subsections below, in
order to build robust, reliable state-of-the-art chemical schemes.

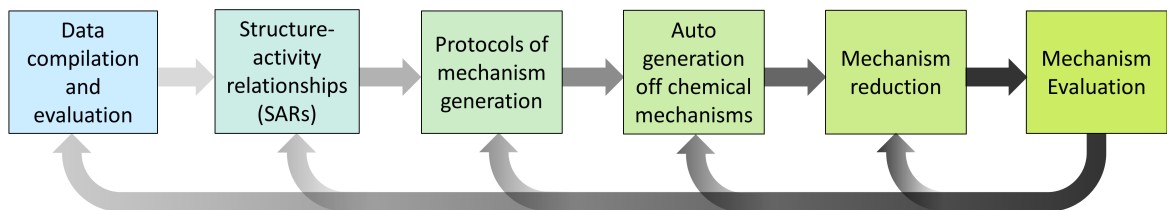

**Figure 3.** Principal steps of chemical mechanism development

## 3.1   Data compilations and evaluations

Over the past approximately 50 years, a rich suite of experimental and theoretical techniques have been developed and employed to provide kinetic and mechanistic data for key gas-phase, heterogeneous, and aqueous-phase processes. This has resulted in extensive atmospheric chemical kinetic databases that contain literature data for several thousand processes, and
continue to grow with approximately a hundred new papers each year. The gas- and solution-phase NIST Chemical Kinetics Databases (Manion et al., 2015), MPI-Mainz UV/VIS Spectral Atlas (Keller-Rudek et al., 2013), and Henry's Law Constants database (Sander, 2023) are online, regularly updated, compendia of reaction kinetics which are extremely valuable. However, they do not provide expert evaluation of the measured data, and thus do not provide recommendations.

In contrast, expert panels have been convened to critically evaluate and provide recommended data for use in atmospheric
science and policy models. Examples include the IUPAC Task Group on Atmospheric Chemical Kinetic Data Evaluation (Mellouki et al., 2021), the NASA Panel for Data Evaluation (Burkholder et al., 2020), and groups that reviewed and evaluated literature data on (i) atmospherically relevant gas-phase reactions of organic compounds available as of 2015 (Calvert et al., 2015), (ii) aqueous-phase kinetic data (Ervens et al., 2003; Herrmann, 2003, 2015; Herrmann et al., 2010) and (iii) atmospherically relevant gas-phase reactions of organic compounds with OH and $NO_3$ radicals, Cl atoms, and ozone (McGillen et al.,
2020). They emphasized that these evaluations are complementary efforts. Such evaluations and the availability of updated fundamental chemical data are the foundation of air quality and climate models and, thus, need to be made as sustainable as possible if we are to make robust and reliable predictions. This is an area that is often overlooked but needs to be funded appropriately.

Aside from the non-trivial task of maintaining and updating these databases, the future challenges for data compilation and
evaluation efforts are to harness the growing power of information technology to gather and assess input data, and to provide recommendations in machine-readable formats to facilitate their use in atmospheric models. As an example of this effort, the IUPAC Task Group is developing machine-readable data sheets to enable automatic updates of chemical mechanisms with the





latest IUPAC recommended data. Such efforts in data evaluation and recommendations are also essential to inform machine-learning applications (Section 4.5) to make their use in future mechanism creation as reliable and valuable as possible.

## 3.2 Structure-activity relationships (SARs)

The development of structure-activity relationships (SARs) to make estimates needed for developing chemical mechanisms for atmospheric models relies on accurate data for the rate coefficients and mechanisms of relevant atmospheric processes. If the underlying chemical mechanisms are understood, it is possible to create SARs to predict them, as it is neither necessary, nor feasible, to measure kinetic rate coefficients and/or branching ratios for all individual reactions.

SAR approaches range from the purely empirical methods that appeal largely to chemical intuition, to those that possess a more theoretical basis. The role of SARs in larger kinetic models, together with emerging needs and current limitations of gas-phase SARs have been discussed extensively, e.g., (Vereecken et al., 2018). The expected increased importance of auto-generation or software-assisted building of more explicit mechanisms will put greater emphasis on reliable and machine-readable SARs to provide kinetic information, including reaction rate coefficients, product distribution and branching ratios as a function of the reaction conditions. This may lead to future challenges in providing SARs for all reaction classes, with a wider scope of applicability that covers the full range of molecular structures in large explicit models. This will also test our ability to validate these SARs to avoid incorrect extrapolation of reactivity trends, and to quantify the uncertainty on the SAR predictions to help in model validation.

In the early 1980s, Atkinson et al. (1982) developed an empirical SAR for estimating rate coefficients that relies on base rate coefficients modified for substituents in the immediate environment of the reactive site. This approach has been widely adopted, together with several variants, owing to its many advantages, including its ease of use, its site-specificity, and the fidelity with which it often reproduces the rate coefficients. This approach is, however, not without its challenges. Multiple functionalities can interact to influence the reactivity non-additively (steric effects, H-bonding, site specific reactivity, etc), and training data to derive higher-order parameters describing these cooperative effects are typically not abundantly available. As a result, large uncertainties still remain in all SARs. Even when a VOC is expected to be oxidized by 'conventional chemistry', automated oxidation mechanisms can have significant cumulative uncertainties in the product spectrum, and SARs often do not work well for even the initial OH abstraction/addition reactions (Kwok and Atkinson, 1995; Jenkin et al., 2018). Novel concepts or strategies may help to tackle this problem to provide workable, though not necessarily simple, SAR models. One such strategy is to incorporate procedurally generated a priori descriptions of substitutions, which may lead to a more robust estimation method, as discussed in (McGillen et al., 2024).

Another area of research is the use of more and/or different molecular descriptors, where pure molecular descriptors (e.g., number of atoms, bonding and positioning of substituents, hydration effects) are supplemented with computational descriptors (e.g., fingerprints, Wiener or Randić indices, topological indices, Fukui indices, wavefunction hardness/softness, or electro-topological state). These latter ones are derived from the molecular wavefunction, typically calculated using an inexpensive methodology (semi-empirical or molecular modeling tools). Another approach that is receiving increased attention is abandoning the traditional SAR setup as a parametrized set of simple equations in favor of more versatile mathematical models





as used in machine learning, such as neural networks. Both these approaches make the resulting SAR harder to interpret by a human, and may even appear as black-box approaches. Therefore, the improved predictive power of such methods needs to be balanced against the potential for inadvertently generating incorrect predictions by extrapolating reactivity trends beyond their scope of applicability.

Contrary to a decade ago, when SARs were primarily developed based on experimental data, current SAR development aims to combine experimental and theoretical data. A particular strength of theoretical studies is that they are well-suited to probe systematic molecular variations; even if absolute accuracy is not achieved, reactivity trends may still be apparent. The use of theoretical studies has opened up new possibilities for the development of SARs for radicals and other unstable intermediates, such as alkoxy and peroxy radicals, or Criegee intermediates. In addition, the use of high-level quantum chemical calculations have been recently extended to derive photolysis data (i.e., absorption cross sections and quantum yields) of atmospherically photolabile species, which have previously been difficult to measure (Prlj et al., 2022; Janoš et al., 2023).

The abundance, variety and role of multifunctional compounds in both the gas and aqueous phases pose another problem for SARs, since there is insufficient training data available to cover the combinatorially high number of possible compounds. The combinatorics also makes systematic studies to derive sufficient training data challenging, both for experimental and theoretical reasons. Novel approaches such as machine learning may help to focus explicit studies to those compounds that have the highest benefit for SAR development.

Reaction rules in the gas phase are often transferable to the aqueous phase, as shown, for example, in correlations of the OH rate coefficients of aqueous- *vs* gas-phase reactions Monod et al. (2005). However, hydration effects and acid dissociation cause deviations from these relationships (Doussin and Monod, 2013). SARs for oxidation reactions of organics are available for other aquatic environments, including those for aromatic compounds by OH, $^1O_2$, and $O_3$ based on the theories by Marcus-Hush and Rehm-Weller (Canonica and Tratnyek, 2003), and for OH reactions with organics derived by machine learning algorithms (Zhong et al., 2021). It seems promising to extend such SARs towards more atmospherically relevant organic species. Accounting for non-ideal (high ionic strength) conditions present in atmospheric aqueous particles poses a specific challenge because most kinetic data are experimentally derived for dilute aqueous solutions. Herrmann (2003) proposed an ion-pairing model accounting for the ionic-strength dependence of rate coefficients. To date, such physicochemically based concepts remain hard to generalize to atmospherically relevant solute mixtures due to the lack of comprehensive data. They should become the focus of future studies to explore the validity of simplifications in multiphase chemistry models that often apply the same kinetic data to both dilute (cloud) and high-ionic strength (aqueous aerosol) solutions. In addition, the mechanisms of organic reactions in aerosol particles, including non-radical and accretion reactions and the role of photosensitizers (Ervens et al., 2011; McNeill et al., 2012), resulting in different products than in (more) dilute cloud droplets are far from being sufficiently understood before rule-based SARs can be developed.

### 3.3 Protocols for mechanism generation

The design of any chemical mechanism relies on a well-defined protocol comprising a set of rules and datasets that govern the construction process. The reliability of these mechanisms rests upon the accuracy and relevance of the prescribed protocol.



Conceptually, the protocol should include sufficient information so that its meticulous and independent application to the same distribution of primary species results in identical mechanisms. It is often the protocol itself that requires regular updates and maintenance to align with the latest understanding of atmospheric chemistry. The protocol must ideally be transparent and accessible, so that the scientific community can actively keep track of and contribute to the maintenance and revision of the

235 mechanisms. The concept of the protocol is not limited to the construction of chemical mechanisms, but also extends to their use. In particular, the mechanisms employed in air quality and chemical transport models often involve the use of model (or surrogate) species that are partially or completely anonymized and lack a direct equivalent in emission inventories (Section 2). As a result, it is imperative to associate these mechanisms with a transparent protocol that precisely outlines how the diversity of emitted species should be adequately distributed among the variables within the mechanism.

240 The concept of protocol has played a pivotal role in the development of detailed mechanisms, particularly in the context of atmospheric oxidation of organic compounds in both the gas and aqueous phases. These mechanisms can be represented by a limited number of reaction types, iterated multiple times until the complete oxidation of the original parent compound is achieved. The inherent redundancy in these reaction sequences enables the design of generic protocols, which facilitate the systematic construction of chemical mechanisms. This approach relies heavily on the utilization of SARs to estimate unknown

245 kinetic, mechanistic or thermodynamic parameters (Section 3.2).

Rigorous application of protocol rules can lead to an unmanageably large number of reactions (Aumont et al., 2005), and practical considerations demand substantial reductions in complexity. These reductions are, for example, achieved through the use of isomer or surrogate species ('lumping') and by selectively excluding certain chemical reactions. Consequently, simplifications and reduction rules have become an integral component of the overall protocol (Section 3.5). The development

250 and application of these protocols have led to various families of mechanisms for gas-phase oxidation of organic compounds, such as the MCM (Jenkin et al., 1997), the SAPRC mechanisms (Carter, 2000, 2010, 2023; Carter and Heo, 2013; Carter et al., 2023) and GECKO-A (Aumont et al., 2005). The same approach has also been proposed and started to be applied for the development of oxidation mechanisms in the atmospheric aqueous phase (Mouchel-Vallon et al., 2013, 2017; Bräuer et al., 2019).

## 255 3.4 Auto generation of chemical mechanisms

The application of protocols for mechanism development has often relied on manual construction, even for detailed mechanisms like the MCM (Jenkin et al., 1997), the NCAR master mechanism (Madronich and Calvert, 1989, 1990; Aumont et al., 2000) and CAPRAM (Herrmann et al., 2000; Ervens et al., 2003). This manual approach is not only time-consuming and subject to human errors, but also severely limits modifications and updates or sensitivity tests. However, more recently,

260 automating mechanism generation tools have been developed, guided by the protocol rules, that systematically and comprehensively describe the reactivity of various compounds. Conceptually, auto-generated mechanisms do not differ from those created manually. The significant advantages of such automated mechanism generation lies in its ability to easily and efficiently create chemical mechanisms to test novel hypotheses or ideas. Moreover, updating a mechanism can be a laborious task, particularly





when the entire mechanism must be revisited due to changes in underlying SARs. In contrast, updating or correcting the coded
SAR rules is relatively straightforward, enabling easy (re)generation of new mechanisms.

Two generators are commonly used for describing the gaseous oxidation of organic compounds: Mechgen for SAPRC mechanism generation (Carter, 2023, 2024b) and GECKO-A for near-explicit mechanism generation (Aumont et al., 2005). These generators can be considered computer tools that emulate the steps of chemical mechanism development, mimicking the steps of mechanism developments by chemists. For a given precursor, generators analyze the chemical structure of the compounds to determine the appropriate reaction pathways. For each identified reaction, rate coefficients and/or products are searched in an experimental database. In the absence of such data, estimates are made based on defined SAR rules. Alternatively, reaction products may be substituted or 'lumped' using predefined surrogate species (as in SAPRC) or existing species within the mechanism (as in GECKO-A). This lumping process also follows a well-defined protocol.

The auto-generation approach enables the development of highly detailed, (semi)explicit oxidation mechanisms, even for parent compounds with complex molecular structures such as monoterpenes, e.g., (Valorso et al., 2011) or long-chain branched aliphatic compounds, e.g., (Aumont et al., 2013; La et al., 2016). These mechanisms encompass hundreds of thousands of species and track the identity and reactivity of organic carbon throughout various stages, leading to final oxidation products. Consequently, these mechanisms may serve as valuable tools for exploring the state and behavior of organic matter during oxidation, examining its impact on pollution episodes, climate, and tropospheric oxidizing capacity. Moreover, the resulting near-explicit mechanisms are rooted in first principles, aiming to reflect the current fundamental understanding of atmospheric transformations. Thus, they provide objective means to quantify our knowledge of atmospheric processes by assessing the disparities between simulation results and observations. This approach offers great potential to investigate the underlying causes of model discrepancies and may inspire the design of new or modified experiments for further testing. However, automatically generated mechanisms are, at least in part, based on relatively crude approximations (e.g., SARs), most of which have significant uncertainties. It is important to undertake sensitivity analyses in order to identify key steps in the reaction pathways, including those where there is a competition between reaction routes.

## 3.5 Mechanism reduction

As mentioned in Section 2, near-explicit mechanisms serve as a benchmark for developing reduced mechanisms, offering a systematic route for designing simpler mechanisms for specific applications. Such an approach enables easy traceability of the simpler mechanism (or parameterizations) back to the reference explicit mechanism, which is, in turn, linked to the current understanding of atmospheric chemistry (Figure 3). Manual application of various reduction protocols have previously been employed to assess their relevance in the gas phase (Aumont et al., 1996; Kirchner, 2005; Utembe et al., 2009) and aqueous phase (Ervens et al., 2003). The reduction process can also be automated in mechanism generation tools, ensuring easy maintenance and traceability of the reduced mechanisms (Szopa et al., 2005; Carter, 2023, 2024b; Carter et al., 2023).

Model simulation results obtained with near explicit mechanisms over a range of conditions can also be used to base the development of simplified mechanisms. The general approach is based on box model simulation results used as a database to train parameterizations. Highly simplified mechanisms can then be developed for organic compounds, in which the products



and their stoichiometric coefficients of a set of pseudo reactions are optimized to best reproduce a particular target, e.g., the evolution of the SOA mass concentration (Lannuque et al., 2018).

Recently, simulation results obtained with a near explicit mechanism were successfully used to train a random forest that emulates the SOA formation and aging (Mouchel-Vallon and Hodzic, 2023). In addition, a reduction tool was designed that iteratively and progressively tests various reduction strategies for organic aerosol mechanisms on an initial detailed benchmark mechanism, based on the elimination of species or reactions and the substitution or lumping of species (Wang et al., 2022). Such approaches may open the possibility of representing the complexity of organic multiphase processes in air quality and

climate models (Woo and McNeill, 2015). However, from a practical point of view, maintaining such mechanisms can present significant challenges due to the necessity of regenerating the training database with each update of the reference mechanism and initiating the development process afresh using the updated database.

### 3.6 Mechanism evaluation

It is essential to continually evaluate chemical mechanisms against theoretical, laboratory, chamber and field data to ensure

accuracy and identify knowledge gaps. Mechanism development has to occur synergistically with developments of instruments and techniques allowing a more stringent testing of mechanisms for a continuously increasing number of species and processes. Evaluations must be done over the relevant ranges of conditions and chemical regimes and for target criteria, e.g., concerning air quality, human or ecosystem health, and/or climate.

The availability of datasets from unexplored or understudied areas led to the identification of new chemical regimes (Section

5.2) for which the applicability of existing chemical mechanisms needs to be carefully evaluated and potentially updated. In certain circumstances, evaluation can be performed based on field studies, but the controlled environments of simulation chambers generally provide a better (and more accessible) testbed (Section 4.3). Improving the representativeness of experimental conditions in lab and chamber studies involves, e.g. lowering the reactant concentrations but also increasing the complexity of the mixtures. For example, SOA experiments have pointed to non-additive effects resulting in lower yields when mixtures of

VOCs are used as compared to single precursor experiments (McFiggans et al., 2019). New measurement techniques, such as proton-transfer-reaction and chemical ionization mass spectrometry (PTR-MS, CI-MS), have led to the identification of multi-generation products and elucidation of chemical mechanisms in aerosol. The direct detection of stabilized Criegee intermediates has allowed for quantitatively studying their kinetics and reaction mechanisms; similarly, speciated $RO_2$ measurements have helped establish the importance of specific precursor VOCs and their oxidation products.

The evaluation of multiphase chemical mechanisms is relatively more difficult and complex. Kinetic data for aqueous-phase processes are usually measured in bulk aqueous phases. However, the atmospheric multiphase system contains dispersed particles or droplets. While the kinetics in the homogeneous gas and aqueous phases are not affected by this physical state, the phase transfer processes and phase states pose major challenges. The sparsity of suitable lab set-ups or chambers for cloud and aqueous aerosol chemistry studies makes it difficult to test multiphase chemical mechanisms under (near) atmospheric

conditions.



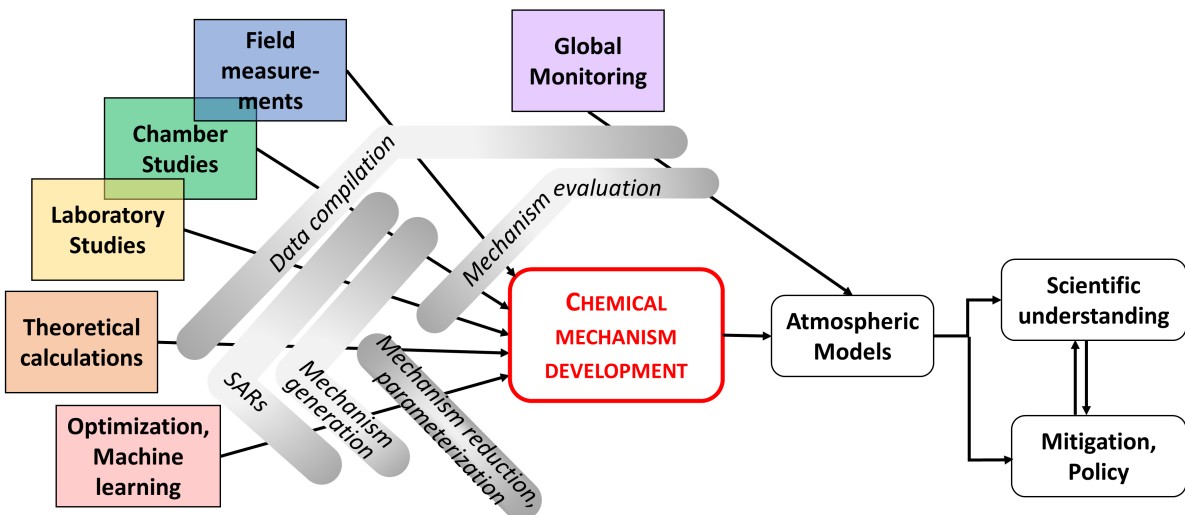

**Figure 4.** Research activities connected to chemical mechanism development for implementation in atmospheric models, leading to an enhanced understanding of atmospheric implications eventually resulting in adequate air quality and climate policies.

## 4  Research activities related to chemical mechanism development

Major advances in measurement techniques and data analysis over the last decades have led to the detection of a multitude of atmospheric chemical compounds, in particular organics, in the gas and condensed phases. At the same time, developments in theoretical, laboratory, field and chamber studies allowed for experiments to derive detailed kinetic and mechanistic data of individual species/products under controlled conditions. Figure 4 illustrates that chemical mechanism developments result from a wide range of insights coming from various research activities, including theoretical studies (ab initio, quantum mechanics) but also field observations, laboratory/chamber studies, and machine learning. It is evident that global monitoring is essential to test the resulting models. Selected aspects and examples of these research activities are discussed in the following subsections; they have triggered the development of new or expansion of existing chemical mechanisms, or have been used to identify gaps in our knowledge that require more research efforts.

### 4.1  Theoretical Calculations

Theoretical calculations can provide significant insights into the mechanisms of specific reactions and can predict the kinetic parameters needed in models. They are highly complementary to experimental studies and provide information for species and/or reaction conditions that are difficult to probe experimentally. Recent developments have led to an increase in the accuracy and range of theoretical calculations for gas-phase reactions, e.g., (Vereecken and Francisco, 2012; Vereecken et al., 2015). From being semi-quantitative at best three decades ago, current high-level ab initio calculations are now able to calculate activation barriers predicted at 'chemical accuracy' (to $\sim$4 kJ mol$^{-1}$ or better) for most systems and agreement between theory and experiment for rate coefficients is often well within a factor of two. Some systems, for which our understanding has been




driven by theoretical studies, include the complex isoprene OH regeneration chemistry, $RO_2$ autoxidation, and the chemistry of
Criegee intermediates. However, not all systems are amenable to precise calculations. Exceptions include calculations involving
decomposition reactions of QOOH radicals (i.e., carbon-centered radicals containing an -OOH group) that are relevant to auto-
oxidation and highly oxygenated organic molecule (HOM) formation as they are subject to significant spin contamination. For
such systems, the agreement between experiment and theory in terms of barrier heights is currently still poor.

Calculations for aqueous-phase or for multiphase processes are likewise complex, as these must describe not only the sur-
rounding matrix (e.g. hydration effect), but also the impact of non-ideality of the aqueous/organic phases and of the interface
with the gas phase. As a result, such calculations are much sparser than for gas-phase processes, and harder to generalize to the
range of atmospheric conditions.

Theoretically calculated data are particularly useful in filling data gaps needed for the development of SARs that are required
for explicit mechanism development. This would be greatly aided if there were a readily available database of evaluated
theoretical results, analogous to what is currently available for experimental results as discussed in Section 3.1. However, such
an effort is not straightforward because of the many levels of theory that can be employed, and the resulting variable reliability
of the results. The rapid advances in accuracy made by theory over a comparatively short period of time requires expertise
to assess the reliability of the results. Any database for theoretical data must, thus, provide ample metadata and evaluation
to avoid interpretation or use of the kinetic data based on incorrect assumptions of their uncertainty interval for the reaction
conditions examined.

## 4.2  Laboratory Studies of Elementary Reactions

Elementary reactions include thermal chemical (uni-, bi- or termolecular) and photolysis reactions. Notable recent develop-
ments in laboratory-based measurement techniques include time and energy resolved photoionization (Osborn et al., 2008),
IR and NIR absorption techniques (Thiébaud and Fittschen, 2006), CI-MS methodologies, multiplexed UV absorption spec-
troscopy (Stone et al., 2018), and frequency comb techniques (Bjork et al., 2016; Roberts et al., 2020). However, conventional
laser-induced fluorescence coupled to laser flash photolysis, discharge flow, and relative rate methods still account for most
kinetic studies of gas- and aqueous-phase reactions. Their application has been extended beyond the traditional measurement
of overall rate coefficients of 'simple' radical reactions with stable molecules (e.g., OH reaction with VOCs), to site-specific
chemistry (McGillen et al., 2016; Onel et al., 2015), for more refined SAR and mechanism developments (Section 3.2). Specific
pathways and mechanisms related to Criegee intermediates and their role in SOA formation have been elucidated (Chhantyal-
Pun et al., 2020; Lin and Chao, 2017; Taatjes, 2017). Despite these recent advances, our understanding of site-specific chemistry
remains limited and further experimental studies should be a priority. Increased efforts to combine theoretical and experimental
studies have contributed immensely to our understanding of isoprene (Berndt et al., 2019a; Fuchs et al., 2013; Medeiros et al.,
2022) and dimethyl sulfide oxidation (Berndt et al., 2019b; Jacob et al., 2024) and needs to be applied more widely to improve
our fundamental understanding of gas- and aqueous-phase reactions.

Photolysis rates remain a challenge to determine in the laboratory, to eventually build SARs, and to incorporate into chemical
mechanisms. Given the key role of these reactions in atmospheric radical cycles, more research is needed in this area. Deter-





mining photolysis rates for atmospherically relevant species requires quantification of absorption cross sections and quantum yields, both of which vary with wavelength, temperature, pressure, and phase (gas, aqueous, heterogeneous). General features

of absorption spectra (cross sections) for organic species are relatively well known, depend on the nature of the chromophore and are often additive for multi-functional species. However, cross sections in the actinic region of the spectrum are often weak (i.e., in the long-wavelength 'tail'), strong functions of temperature, difficult to measure, can depend on substituents to an uncertain extent, and very difficult to predict. Some quantum yield data are relatively simple to measure or predict, for example, values of unity for organic hydroperoxides and nitrates, due to the nature of the electronic state being accessed and the

presence of sufficiently weak bonds. In contrast, photolysis of the atmospherically abundant carbonyl-containing compounds can be far more complex. Multiple photolysis pathways, collisional quenching, intersystem crossing, and internal conversion make determination of photodissociation quantum yields over the full parameter space challenging, and predictions difficult or untenable.

Similar to kinetic rate coefficients in the condensed phases, photolysis rates also often show similar trends to the better
constrained data for gas-phase processes. In addition, to such direct photolysis processes, the photosensitizers in and on aqueous particles represent an additional challenge (Bernard et al., 2016; Malecha and Nizkorodov, 2017; Mekic et al., 2019; Felber et al., 2020). More comprehensive studies are warranted to systematically implement these processes into multiphase chemistry models to assess the importance of photosensitizers under different conditions.

### 4.3 Atmospheric Simulation Chambers

Atmospheric simulation chambers provide a compromise between the tightly controlled conditions in laboratory experiments and the real atmosphere, and are also used as excellent testbeds for the development, intercomparison and calibration of instrumentation. The European PHOtoREactor (EUPHORE) in Valencia can be considered a pioneering outdoor chamber facility, which, along with more recent large outdoor chambers (e.g., SAPHIR at Forschungszentrum Jülich) allow for decreasing the surface-to-volume ratios to minimize chamber wall effects, in conjunction with sensitive detection methods, for long duration

studies under close-to-ambient concentrations (Rohrer et al., 2005; Jeffries et al., 2013; Li et al., 2021). The huge improvement in the range and sensitivity of chamber instrumentation has led to the ability to comprehensively monitor the chemical evolution of precursor and product species, including radical intermediates such as $HO_2$ and $RO_2$ (Fuchs et al., 2008; Onel et al., 2017) and OH reactivity (Fuchs et al., 2017).

During the past 20 years, the development of a broad range of indoor and outdoor atmospheric simulation chambers has
facilitated many fruitful collaborative efforts, that also include making chambers accessible to the community. The European EUROCHAMP network (https://data.eurochamp.org/), has done an excellent job in promoting and applying best practice in terms of developments of new techniques and approaches, creation/curation of publicly available databases and making European chamber infrastructure more integrated, sustainable and accessible (Doussin et al., 2023). These efforts are sustained under the ACTRIS umbrella (https://www.actris.eu/facilities/national-facilities/exploratory-platforms).

The availability and development of chambers for multiphase systems is not as advanced as those for gas-phase chambers. While numerous cloud chambers have been available for some time (e.g., CESAM (https://cesam.cnrs.fr/); AIDA (https:





//www.eurochamp.org/simulation-chambers/AIDA), these are mostly equipped to explore (micro)physical processes and/or aerosol-cloud interactions, or are dedicated to aerosol chemistry experiments under non-cloud conditions. Cloud chemistry chambers would likely yield important new insights into the complex multiphase chemistry system (Section 5.4). Data centers, including the EUROCHAMP/ACTRIS ASC (https://www.actris.eu/topical-centre/data-centre), give easy and sustainable access to experimental and modeled data obtained from simulation chamber experiments, as well as resources, protocols, mature products and tools that support activities related to measurements and modeling. More recently, the ICARUS data center has been developed to give access to data from US simulation chambers (https://icarus.ucdavis.edu/).

## 4.4 Field Studies and global monitoring

Data from field studies are the ultimate test to confirm the accuracy and completeness of chemical mechanisms. As with chamber studies, field studies in a broad range of locations, from highly polluted megacities to the remote, pristine rainforest, have advanced enormously due to instrument developments, evaluation and intercomparisons. They have led to immense data sets from long- and short-term experiments and monitoring and include sensitive radical intermediate and reactivity observations; similarly to the wealth of data from chamber experiments. Such valuable datasets require consistent data processing, curating and archiving, and transparent and open provision to the community. Field measurements on the ground and/or by aircraft may be complemented by remote sensing data. Satellites and sensor networks can provide concentration data with wide spatial (global) coverage. Given their limitations in terms of species, resolution and specificity, their immediate role in informing mechanism development efforts seem currently limited. However, new and emerging satellite products and space borne platforms are coming online and the sensitivity and retrieval coverage are improving all the time (Wells et al., 2020).

In addition to complementing existing datasets on previously investigated locations, a large focus of field studies in recent years has been to study locations with particular chemical characteristics (Section 5. The growth of urban agglomerations into megacities or even a gigacity (Kulmala et al., 2021) have significant consequences for air quality, human health and climate. The immediate impact of atmospheric processes on air quality and human well-being in such areas motivated numerous field campaigns in densely populated regions, e.g., MILAGRO, 2006 (Molina et al., 2010), UK APHH Programmes in Beijing (Shi et al., 2019) and Delhi (Nelson et al., 2021). Extreme chemical conditions leading to a suppression of NO due to very high $O_3$ levels and hence a change in $RO_2$ chemistry (Zou et al., 2015; Newland et al., 2021) have led to the definition of new chemical regimes or even the questioning of their concept in general (Wennberg, 2024) that require the adjustment of existing or development of new chemical mechanisms (Section 5.2). Anthropogenic climate change and the implementation of net zero policies is driving significant changes in emissions and atmospheric composition. Sustained and expanded long-term ambient observations from a range of platforms and locations are needed to track the evolution and atmospheric composition, and to develop and test our chemical understanding.

## 4.5 Machine learning

Chemical mechanism development, like many chemistry and physics fields, has greatly benefited from advances in machine learning; at the same time, the rapid developments in machine learning pose a risk, since the underlying assumptions for the



underlying algorithms and their evolution are not fully transparent. Machine learning has been successfully applied to building chemical mechanisms (Sanches-Neto et al., 2022), and to predict partitioning coefficients between the gas and the (organic) particle phases (Lumiaro et al., 2021). A combination of quantum chemical mechanisms and machine learning is a promising approach to quantitatively predict the chemical pollutant degradation by homogeneous or heterogeneous processes (Xia et al., 2022). In addition to the creation of chemical mechanisms, machine learning has been applied to reduce chemical schemes and

develop surrogate models to limit the huge computational burden of explicit chemical mechanisms for the gas and aqueous phases (Kelp et al., 2022; Berkemeier et al., 2023). Shen et al. (2022) significantly reduced the computational costs of a global model with an adaptive algorithm splitting the full chemical scheme into sub-mechanisms.

More recently, graph theory has been shown to be a powerful tool for the investigation and interpretation of complex chemical systems (Silva et al., 2021; Wiser et al., 2023; Sander, 2024). These tools visualize chemical mechanisms and illustrate the

relative importance of individual pathways depending on conditions, which ultimately allows the removal of unimportant pathways to result in smaller mechanisms.

Overall, such tools are certainly a step towards the efficient development of explicit and reduced or empirical chemical mechanisms. While their use seems very tempting, their application as a 'magic black box' should be cautioned: as for any chemical mechanism, the predictions are most robust and certain for parameter ranges that are covered by the training data. In

analogy to data reporting for experimental studies, algorithms from machine learning and published results thereof should be documented together with the conditions and limitations for which they were derived. Algorithms that do not allow reconciling or tracing back individual chemical processes or species might be a useful tool to quickly predict chemical concentrations in well-defined chemical regimes; however, they clearly bear the risk of uninformed usage and thus biased predictions beyond their validity limits. Their application should therefore always be accompanied by more explicit models for comparison and

testing. Generally, all training data sets should be transparently, continuously and interactively updated based on findings from the complementary research activities summarized in Section 4.

## 5   Current and future challenges

### 5.1   New insights into VOC oxidation

Central and well-studied components of chemical mechanisms such as peroxy radical ($RO_2$) reactions as intermediate steps in

VOC oxidation and SOA formation continue to trigger new insights and subsequent research efforts. Until recently, the fate of ($RO_2$) radicals in the gas phase was assumed to be dominated by bimolecular reactions with $NO_x$, $HO_2$, $NO_3$ (at night) and other $RO_2$ radicals. However, recent studies have demonstrated the importance of hydrogen-shift reactions, which have been implicated in HOM formation (Bianchi et al., 2019). The rates of these unimolecular reactions, and their competitiveness with bimolecular reactions are highly dependent upon molecular structure. The significance of such unimolecular (autoxidation)

pathways, for a subset of $RO_2$, was prompted by measurements of large OH concentrations above tropical forests (Lelieveld et al., 2008), which initiated successful collaboration and interactions between laboratory, theory, field and mechanism development activities to further elucidate isoprene oxidation chemistry.




Studies in remote regions, urban areas where high ozone levels efficiently titrate NO to NO2, or during the night have led to an increasing need to understand the rate coefficients and mechanisms for other bimolecular reactions of RO$_2$. These

additional RO$_2$ losses under 'low-NO' conditions include the unexpectedly high reactivity of certain combinations of RO$_2$ and alkene reagents under ambient conditions (Nozière et al., 2023), the formation of stable ester and ether accretion products from RO$_2$ self- and cross-reactions (Peräkylä et al., 2023), and the formation of dimers from RO$_2$+RO$_2$ recombinations (Bates et al., 2022). For some biogenic VOCs, SOA yields were observed to vary from 0 to >80%, depending on the dominant RO$_2$ reaction pathway. This strongly highlights the need of experiments for various and more representative RO$_2$ fates, and also

to implement the various loss pathways into chemical mechanisms to more accurately interpret obtained data across chemical regimes. Criegee intermediates, formed in the ozonolysis of unsaturated VOCs, represent a new class of oxidants with many branching pathways that have been identified only recently. As with RO$_2$, unimolecular hydrogen-shifts are important, as are ring-closure reactions and decomposition, each of which depend strongly upon conformer and substitution patterns (Vereecken et al., 2017).

Furthermore, there are a wide and varied range of bimolecular reactions in which Criegee intermediates may also participate. From a mechanistic perspective, the insertion reactions that occur with a broad variety of labile hydrogen-containing compounds (McGillen et al., 2017; Chhantyal-Pun et al., 2019) are of significance since these may lead to a complex matrix of functionalized peroxidic products whose chemistry remains highly uncertain. The intricacies of gas-phase ozonolysis chemistry can be attributed to the exothermicity of the initiation step, coupled with a complex landscape of weakly bound intermediates

and local potential energy minima. Ensuing competition between unimolecular processes, collisional energy transfer and bimolecular reactions results in a complex product distribution. This was recently demonstrated in the unexpected formation of extremely long-lived products from the ozonolysis of hydrofluoroolefins (McGillen et al., 2023). Consequently, we anticipate ozonolysis chemistry being a continuing topic of study into the foreseeable future.

While it seems reasonable to assume that the similar pathways also occur in the aqueous phase, much less data and mech-

anistic insight into such pathways is available there. RO$_2$ recombinations and other (organic) radical reactions are likely to initiate the formation of high molecular weight compounds (oligomers, accretion products) that have been identified in field and lab studies as SOA compounds. While oligomer formation in laboratory experiments was shown to be consistent with data on oligo/polymerization for isoprene oxidation products in general (Ervens, 2015), these pathways cannot be generalized yet for implementation into detailed chemical mechanisms.

## 5.2 New chemical regimes

Recognition of a vast array of atmospheric chemical challenges over recent decades has expanded our horizons to include chemical regimes of more extreme conditions, such as high temperatures in wildfire plumes and low pressures in the free troposphere, but also over large changes in levels and ratios of key pollutants (e.g. NO$_x$, mobile combustion sources). However, mechanisms that were developed and/or optimized for a specific chemical regime may not be applicable - possibly even mis-

leading - in such regimes. Specifically, the applicability of traditional chemical mechanisms depends on whether the oxidation




processes are occurring via established VOC chemistry or whether new chemistry needs to be described, either because the VOC has novel functional groups or the conditions are novel.

The emergence of new chemical regimes often arises from changing emission profiles. As vehicular emission standards become more stringent and $NO_x$ concentrations continue to decline, urban areas exhibit lower $NO_x$ levels. The resulting new
mixtures of VOC, $NO_x$ and particles may even represent a new 'aerosol-inhibited regime', in addition to the more conventional $NO_x$ and VOC-limited regimes (Ivatt et al., 2022). In this new regime, the $HO_2$ loss is largely controlled by reactive uptake onto aerosol particles, which highlights the need for careful studies of its uptake coefficient for wide ranges of particle composition. As a further consequence of changes in emission patterns, the acidity of cloud water may be increasingly controlled by organic acids (Lawrence et al., 2023, 2024) which are often neglected in thermodynamic pH calculations.

Megacities are growing in size and number and are environments where established mechanisms for oxidation of known VOCs may not be appropriate. Very high $O_3$ concentrations lead to a suppression of $NO_x$ and the reaction pathways for $RO_2$ species may be very different from those of the more traditional high or low $NO_x$ environments (Section 5.2). As global change is evolving resulting in a warmer, dryer climate, increasing occurrences of wildfires in populated areas represent new challenges in terms of air pollution as they affect local and regional air quality through direct emission of a large set of pollutants.
These plumes contain products of incomplete combustion processes of biogenic compounds that are rapidly processed under extreme conditions in terms of temperature and concentration levels, leading to the formation of a wide range of secondary pollutants both in the gas and condensed phases. The highly dynamic nature of such plumes make measurements at sufficiently high temporal and spatial resolution difficult which then translates into the development and testing of suitable chemical mechanisms. To better constrain the chemical processes occurring in such scenarios, controlled experiments are needed to
systematically generate and iterate suitable mechanisms.

### 5.3 New chemical compounds

'New' atmospheric chemical compounds can be defined in multiple ways: Chemical compounds may have been in the atmosphere for a long time but their presence is *new to atmospheric science* and have only been detected recently thanks to advanced measurement techniques and/or based on evidence from lab or theoretical studies or model-measurement discrepancies. The
latter demonstrated that a significant fraction of ambient OH reactivity is unaccounted for which makes predictions of the overall reactivity and oxidation capacity, and ultimately self-cleansing efficiency, of the atmosphere uncertain. Some of this 'missing' reactivity may be due to heterogeneous processes or inorganic reactions, but it is likely that a significant fraction is due to reactions with VOC compounds, either primary emissions or secondary products, that have not been measured or identified yet and thus are missing in current chemical mechanisms. They include increasing relative contributions of non-
vehicular industrial and consumer volatile chemical products (VCPs) (McDonald et al., 2018) and a multitude of higher carbon number VOCs that are becoming detectable by new and innovative methods such as multi-dimensional chromatography (Lewis et al., 2000; Dunmore et al., 2015) and various forms of chemical ionization mass spectrometry. Once they are identified, the protocols for mechanism generation can be applied (Section 3.3).



A second group of new species requiring mechanism development arise from their introduction into the environment via anthropogenic activities. Such compounds include replacements for other chemicals, such as industrially important fluoro-chemicals (hydrofluoroolefins, hydrofluoroethers, fluorotelomers, etc.) (Michelat et al., 2022; Tokuhashi et al., 2018). The environmental fate and persistence of their oxidation is a potential concern, especially the formation of per- and polyfluo-roalkyl substances (PFAS) (Kwiatkowski et al., 2020). Detailed atmospheric chemical models are usually not configured for halogenated species, which limits attempts to understand PFAS chemical interactions and degradation in the environment. Thus, for such compounds the full suite of steps as outlined in Figure 3 need to be followed to generate robust and operable mechanisms to be integrated in atmospheric models.

Finally, climate change mitigation may lead to chemical interactions in the atmosphere that are not captured in current mechanisms and, thus, require further study. Examples include amine oxidation chemistry from large scale introduction of amine-based carbon capture and storage (CCS) schemes and heterogeneous chemistry associated with possible geo-engineering schemes to reduce the solar radiation flux. Chemical processing of amine emissions occurs both in the gas and aqueous phases (Nielsen et al., 2012; Karl et al., 2014), and considerations for multiphase systems (Section 5.4) need to be applied to assess impacts of increasing amine concentrations in the atmosphere.

## 5.4 New considerations on multiphase systems

Intercomparisons of multiphase chemistry models revealed large discrepancies in predicted cloud droplet number concentration and radiative forcing due to in-cloud sulfate formation (Kreidenweis et al., 2003) and in predicted oxidant concentrations in both phases (Barth et al., 2021). The models participating in these exercises only differed in the chemical mechanisms, i.e., the set of kinetic and mechanistic data for gas, aqueous and phase transfer processes. Detailed multiphase chemistry models have pointed to the importance of considering drop-size resolved chemical composition, in particular acidity, in affecting sulfate (Barth, 2006) and the phase partitioning of chemical compounds that affects their atmospheric residence time (Tilgner et al., 2021). Furthermore, cloud droplet size may affect the rates of uptake and chemical conversion within the aqueous phase (McVay and Ervens, 2017; Ervens, 2022). Such sensitivity studies demonstrate the challenges associated with model simulations of the atmospheric multiphase system. While robust and evaluated chemical mechanisms are a basic criterion for such models, the dependence of the reaction rates as a function of the chemical composition and microphysical properties of cloud droplets adds another layer of complexity and uncertainty in prediction of the atmospheric oxidative capacity and aerosol radiative forcing. These considerations suggest that parameterizations based on bulk properties (e.g., total cloud liquid water content or monodisperse drop populations) as often applied in large-scale models, may not be sufficient for all chemical systems. Machine learning approaches seem well suited to further map wide parameter spaces of the multiphase systems and their feedback on chemical reaction rates. This does not only apply to (relatively) dilute cloud droplets but also to less dilute, possibly viscous and/or semi-solid aerosol particles with limited penetration depth (Shiraiwa and Pöschl, 2021; Hua et al., 2022).

The formation and processing of the organic particle fraction that absorbs light (brown carbon (BrC)) poses another par-ticular challenge since it comprises a small percentage (∼1%) of the organic particle mass. The light absorption is attributed





to only a very small number of individual molecules. As a result, developing explicit chemical mechanisms for their description is challenging. Instead, an empirical approach parameterizing the mass absorption efficiency of ambient BrC may be a
more feasible starting point that could be successively expanded to include more information once it becomes available from complementary studies.

In addition to their effects on climate, the role of aerosol particles affecting human health have been clearly demonstrated. The main criterion for adverse health effects is thought to be the total $PM_{2.5}$ mass. However, enhanced occurrence of mortality due to cardiorespiratory diseases in the presence of biogenic SOA may point to specific precursors or reaction pathways (Pye
et al., 2021) that trigger adverse health effects due to reactive oxygen species (ROS) in aerosol particles (Shiraiwa et al., 2012). Combining such empirical trends in adverse health effects with knowledge on explicit ROS reactions in multiphase systems may be a promising avenue to connect atmospheric chemical processes to (physiological, health-related) target parameters.

### 5.5 FAIR models and toolchains

To perform chemical process modeling, the chemical mechanism needs to be converted to differential equations and run
through a suitable integrator. This is done by software able to import a specific notation for the chemical reactions (e.g. Facsimile, KPP, or Gecko-A) and the equations for the rate coefficients (e.g. Fortran, C, Matlab, or parameters for a fixed set of equations, together with support for variables, functions, and other implementation aides). These numerical integrators, in turn, are typically embedded in a much larger software tool chain that also provides support for the intended use of the kinetic modeling such as importing experimental data, providing fitting routines, handling time-specific photolysis rates, exporting
and transforming modeling results, visualization, statistical analysis, integration in an Earth system model, etc., where each of these is implemented differently depending on the task at hand. Furthermore, the various models use different chemical notation for their mechanisms, e.g. isoprene is "C5H8" in the MCM, "CH3Cd(=CdH2)CdH=CdH2" in Gecko-A, "CH2=CH-C(=CH2)-CH3" in MechGen, "C=C(C)C=C" in SMILES notation, and possibly "ISOP" in a locally developed model.

Transferring a chemical mechanism from one tool chain to another is less than obvious, and even more so if one would like
to combine (parts or) models, e.g., start with the MCM, add chemical reactions based on theoretical calculations, and complete with Gecko-A-generated chemical mechanisms for new reaction products. To date, information exchange between models relies all too often on laborious and error-prone manual effort. This opinion paper lists many aspects, for which we need even more elements in the tool chain, such as model reduction, implementation/application/adaptation of SARs, model generation, visualization, evaluation, optimization. It is unrealistic to hope that all tool chains will implement these aspects independently,
and even less so that all applications will converge on a single toolchain. Indeed, the many tool chains exist because of the different needs in various applications, e.g. the spreadsheet-based input for Kintecus is excellent to teach students benefiting from a familiar interface, but would be a very poor choice in global Earth system modeling where the chemical module is only a small part of the performance-optimized computations.

The scientific community is moving towards free, open, and FAIR data (**F**indable, **A**ccessible/Annotated, **I**nteroperable,
**R**eusable, https://www.go-fair.org/fair-principles/), and many journals and funding agencies now demand that data, models, and software are FAIR. Atmospheric mechanisms are usually free, open and *Findable*, as they are described in the literature.





They are, however, less *Accessible*, often relying on cryptic notations for molecules without a user-friendly browsing tool. The MCM website (mcm.york.ac.uk) is an example of a more human-oriented interface, but all models would benefit from having both human-oriented molecular representations as well as unambiguous, traceable, machine-readable notations for
the molecules, all available in a public vocabulary, and consistent for gas and aqueous phases. The literature and protocol papers provide annotation and provenance for the individual models, but once models are merged, modified, or reduced, this information is ill-matched. Provenance should be traced at the level of the individual reactions, but is rarely implemented (e.g., CAABA/MECCA supports adding literature references to specific reactions).

The various tool chains, and thus the models that are geared towards a single chain (subset), are essentially non-Interoperable
and the models are therefore also not generally *Interoperable* or *Reusable* without significant effort. Without interoperability, the ability to rigorously validate modeling runs is lost. One of the challenges for implementing the many aspects discussed in the current article thus also lies in providing tools to make any solutions available and transparent to the community without excessive duplicate effort across the toolchains, while maintaining transparency for the underlying data and the model build.

## 6  Conclusions and future research needs

This opinion article provides a brief summary of some of the developments and gaps that define current and future efforts in chemical mechanism development. It is no coincidence that a large number of references cited here have been published in *Atmospheric Chemistry and Physics* that covers atmospheric chemical studies of a broad range of research activities, including fundamental, lab, field, chamber and machine-learning studies that form the cornerstones for development of chemical mechanisms. While historically, gas phase chemical mechanism developments usually followed one of two strategies, i.e. ei-
ther explicit mechanisms based on fundamental parameters and processes, or empirical, operationally defined mechanisms, respectively, we propose that these approaches are not mutually exclusive: different levels of chemical complexity are required to answer the breadth of scientific questions over a range of scales. The wealth of chemical information provided by new analytical instrumentation and by cheminformatics methods, including machine learning, data mining and assimilation, offers opportunities for bridging the approaches, i.e., supplementing empirical approaches with (nearly) explicit information.

The last two decades have seen huge developments in analytical techniques and instrumentation that have allowed for the detection of more detailed speciated compounds in lab, field and chamber studies. New state-of-the-art measurement techniques have led to the identification of multi-generational products and elucidation of chemical mechanisms not only in the gas phase but also in the condensed phases (cloud droplets, aerosol particles). These new experimental techniques have helped immensely in evaluating and pinning down key processes and knowledge gaps in our understanding of the underlying processes. Continued
development of sophisticated instrumentation, in particular to measure the vast suite of partially oxidized, multi-functional organic species has been and will remain critical. Branching ratios or site-specific rate coefficients, are particularly challenging to obtain. Developments in the general area of mass spectrometry, including arrays of chemical ionization schemes ($H_3O^+$, $NH_4^+$, $I^-$, $NO_3^-$, $CF_3O^-$, etc.), time-of-flight and Orbitrap techniques have been extremely beneficial in this regard. However,




calibration of these techniques for the enormous range of multifunctional species is a daunting task that must be dealt with.
This requires strong reliance on synthetic chemists to provide authentic standards of (at least) representative species.

Any research efforts in these directions should also extend to the characterization of processes in and on cloud droplets and particles. Elucidating the chemical mechanisms in the atmospheric aqueous (and possibly organic) phase in modifying the chemical composition of aerosol particles is essential, as these processes affect aerosol properties, which in turn affect radiative forcing, and human and ecosystem health. As such, the (physico)chemical feedbacks and interactions between multiple phases
must be thoruhgly accounted for in mechanism development. To date, the mechanistic understanding and developments for chemical mechanisms in the condensed phases significantly lag behind those in the gas phase. Specifically, such developments are suffering from the higher complexity of the multiphase chemistry system that may also require the consideration of additional physicochemical (e.g. ionic strength) and microphysical (e.g., size of dispersed droplets or particles) parameters to comprehensively and reliably predict chemical formation, processing and degradation.

Rapid increases in computational power have opened new avenues to develop tools and approaches to complement or partially replace time consuming, tedious, error-prone manual mechanism generation. Such automated techniques are doubtlessly extremely useful tools for mechanism development, and also for the creation of reduced mechanisms to address a variety of research questions and to inform policy makers. Despite rapid progress in theoretical calculations, experimental studies are required more than ever to complement theory and to inform mechanism development efforts, either by traditional 'manual'
mechanism generation or automated tools. However, parameterizations and/or empirical (surrogate) models derived by theoretical and machine learning approaches bear the risk that they are used as 'black boxes' since the underlying algorithms are not fully transparent. Therefore, we call for similar thoroughness and guidelines in reporting the conditions for which algorithms were trained and their limitations. Clear communication of the validity ranges of chemical mechanisms, and specifically the SARs used to build them, is even more important since the needs became evident to extend current mechanisms to conditions
of new chemical regimes (e.g. mega/giga cities, wildfires) that emerged due to societal developments during the last decades and into the future, as the climate continues to change.

In addition to focusing on such new scenarios, it is also particularly important that research funding agencies look sympathetically and are supportive on applications of new techniques that look to re-visit 'established' chemical systems and parameters (e.g. rate coefficients, photolysis cross sections and quantum yields, vapor pressures, SAR development) that are
fundamentally important to atmospheric chemistry. New insights from experimental and theoretical work have revealed significant uncertainties in important inorganic reactions (e.g., $OH + NO_2$ that are relevant to oxidant levels and, in turn, for atmospheric oxidation chemistry in general. The advent of new experimental techniques, such as frequency combs, facilitates the study of these foundation reactions since they can identify potential systematic errors in previous studies and/or improve the precision of rate coefficients.

The vast number of organic species and their pathways leading to highly oxidized and multifunctional products represents a particular challenge for atmospheric chemistry. The reactivity of these products cannot be accurately described by traditional SARs, which calls for dedicated studies to unravel the complexities to ultimately develop suitable SARs and chemical mecha-



nisms. Otherwise, applying insufficiently constrained SARs will propagate into large errors in the prediction of the reactivity and concentration levels of organics and their potential to form SOA.

The most fundamental requirement to create and further develop/update chemical mechanisms is the availability and access to reliable data, along with the associated metadata, from experiments and observations, as enabled by data centers. Careful data curation and evaluation by experts should be supported accordingly. Ensuring free and open access to such data, following FAIR (Findable Accessible Interoperable Reusable) principles must become the standard in our scientific community. Furthermore, publishing open codes for modeling studies, as already mandatory in the open-access EGU/Copernicus journal *Geoscientific*
*Model Development* and strongly encouraged in all EGU journals, should become the standard requirement to make mechanism creation and application transparent and accessible.

    Research efforts combining lab, field, chamber, theory and model studies, in which the chemical mechanism is at the core, have been shown to successfully lead to the mitigation of significant environmental issues (e.g. acid rain, stratospheric ozone depletion) demonstrating the success of synergistic open research efforts of the atmospheric chemistry community. It remains
critical that we maintain such research capacity, both in terms of experimental and theoretical studies, and 'blue skies' funding programs to pre-empt potential global problems and to identify and respond rapidly to emerging issues.

*Data availability.*    No new data were generated for the article. All relevant data can be found in the cited literature.

*Author contributions.*    AR and BE contributed equally to the development of the conceptual framework for this research and coordinated the individual contributions written by the other coauthors.

*Competing interests.*    At least one of the (co-)authors is a member of the editorial board of Atmospheric Chemistry and Physics.

*Acknowledgements.*    We gratefully acknowledge the Coordinating Research Council's Atmospheric Impacts Committee who have provided valued support to the authors work on the development and evaluation of databases and estimation methods for predicting air quality impacts of emitted organic compounds, and we appreciate our fruitful discussions with Sasha Madronich (National Center for Atmospheric Research).



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
