# Peer review of "Opinion: Challenges and needs of tropospheric chemical mechanism development"

_EGUsphere, 2024_

## Author Comment (AC1)

**Author response to referee and community comments**

We thank John Wenger and two anonymous colleagues for their thoughtful referee comments, and Mike Jenkin for his useful community comment. All comments are shown in black, our responses in blue; manuscript text is indicated in **red** with additions in **bold**.

**Referee #1 (John Wenger)**

This opinion paper provides an excellent overview of the challenges and needs for the continued development of tropospheric chemical mechanisms to support research and policy on air quality and climate. Following a summary of the strategies and procedures currently employed in chemical mechanism development, the authors cover key advances in the experimental and theoretical research activities that are used as the basis for developing and improving the mechanisms. The current and future challenges for chemical mechanism development are well considered and in general, clearly presented. The authors finish up with some strong conclusions on research needs for the future.

The article is written by key experts currently working in chemical mechanism development and associated research fields and, as such, provides valuable and insightful commentary on the state-of-the-art, challenges and future needs in chemical mechanism development. In general, the article is well-constructed, has good information flow and the errors/typos are at a minimum. I am in favour of publication following appropriate responses to the minor comments below.

Page 2, lines 17-21: The phrasing in these two sentences could be improved. Change "photochemical degradation" to "atmospheric degradation" as not all oxidation processes are photochemical.

We changed the text as follows (l. 17):

**The atmospheric degradation of these compounds is complex and nonlinear and results in the formation of a wide range of potentially harmful secondary pollutants including ozone and secondary organic aerosol (SOA). Understanding this chemistry requires numerical models with chemical mechanisms that include kinetic and mechanistic information describing the formation, cycling and degradation of chemical compounds.**

Page 2, line 29: Oxidants such as. . . . . . .

We assume that the referee refers to line 39. We extended the sentence as follows (l. 37):

The role of multiphase chemistry for chemical budgets in the atmosphere has since then been further explored and expanded towards the inclusion of oxidants **(e.g., reactive oxygen species, epoxides, $^1O_2$)** , and organic compounds

Page 3, line 54: Complementarity instead of complementary

Thanks. We replaced 'complementary' by 'complementarity'

Page 3, lines 55-59: It appears that section 3 and section 4 have been mixed up here

The referee is right. Thanks for pointing this out. We flipped the sentences and changed the numbers accordingly (l. 57):

In Section  **3**, we systematically outline the various steps needed to develop chemical mechanisms ensuring their reliability and robustness, together with an overview of progress and status for each individual step. The spectrum of research activities in terms of theoretical and experimental studies that are required to inform mechanism developers of reliable fundamental data is described in Section  **4**, briefly summarizing recent advancements, current status and perspectives.

Page 3, line 62, title: Strategies for chemical mechanism development

We replaced the section title 'Strategies of chemical mechanism development' by 'Strategies for chemical mechanism development' as suggested by the referee.

Figure 1: In the green box, replace "constants" with "coefficients"

We replaced 'constants' by 'coefficients'

[Figure]

Figure 1: Outline of approach for development of (semi)explicit mechanisms to be used primarily to compile scientific knowledge or interpret scientific data.

Figure 2: I really like this figure!

Thanks!

Page 5, line 98: . . . continued funding or continuing to fund. . .

We replaced 'continuing funding' by 'continued funding'

Page 5, line 104: environmental chamber or atmospheric simulation chamber?

We replaced 'environmental chamber' by 'atmospheric simulation chamber'

Page 7, line 138, title: Steps involved in chemical mechanism development or General procedure for chemical mechanism development

We replaced the section title by 'Steps involved in chemical mechanism development'

Figure 3, caption: Principal steps involved in (or General procedure for) chemical mechanism development

We replaced the figure caption by 'Principal steps involved in chemical mechanism development'

Page 7, line 155: the sentence "They emphasized. . . ." Should be rephrased and possibly moved to somewhere else in the paragraph. Also, here the authors might want to strengthen the last sentence to give it more impact.

We agree with the reviewer's comment and have modified the last three sentences of the paragraph to read (l. 159):

**Compiling and evaluating data are complementary efforts. Evaluated fundamental chemical data are the foundation of all air quality and climate models and need to be updated on an ongoing basis to ensure that models are based on the latest science. Data evaluation is critically important for continued progress in the field of atmospheric chemistry and for the development of effective policy and regulations to address air pollution and climate change. As such, data compilation and evaluation need to be funded accordingly.**

Page 8, line 170: . . . that rely largely on chemical intuition,

We replaced '. . . that appeal largely to chemical intuition' by '. . . that rely largely on chemical intuition' as suggested.

Page 10, line 240: "The concept of protocol. . . ." sounds a bit odd. Consider re-phrasing.

We changed the text as follows (l. 245):

 **Protocols have** played a pivotal role in the development of detailed mechanisms . . .

Page 10, line 255, title: Replace "Auto" with automated or automatic, whichever is more appropriate

We changed the section title as follows:

3.4 Auto**mated** generation of chemical mechanisms

Page 10, line 260: Is "automating" correct here?

We have replaced "automating" with "automated". (l. 264)

Page 11, 268-269: suggest. . . emulate the process of mechanism development carried out by chemists

We changed the text as follows (l. 273):

These generators can be considered computer tools that emulate the  **process** of chemical mechanism development **as it is carried out** by chemists.

Page 11, line 276: might want to add the huge number of chemical reactions for emphasis

Thanks for the suggestion. We added some numbers as follows (l. 281):

The auto-generation approach enables the development of highly detailed, (semi)explicit oxidation mechanisms, even for parent compounds with complex molecular structures such as monoterpenes, e.g., (Valorso et al., 2011) or long-chain branched aliphatic compounds (Aumont et al., 2013; La et al., 2016). **For instance, in the case of C7 alkanes, this may result in $10^5$ species and $10^6$ reactions (Aumont et al., 2005).**

Page 11, line 282: might want to specify model results or model simulation results to avoid any potential confusion with simulation chamber experiments

We clarified it as follows (l. 287):

. . . , they provide objective means to quantify our knowledge of atmospheric processes by assessing the disparities between  **model** results and observations

Page 12, line 300: might want to specify model results or model simulation results

We clarified it as follows (l. 311):

. . . ,  **model** results obtained with a near explicit mechanism were successfully used

Page 12, line 304: Current phrasing implies that organic multiphase processes are not represented at all. Is this true? If not, then the phrase "may lead to more accurate representation of the complexity" more appropriate here.

This meaning was actually not what the authors intended to convey. We therefore changed the sentence as follows (l. 315):

Such approaches may  **allow a more accurate representation of the** complexity of organic multiphase processes in air quality and climate models (Woo and McNeill, 2015).

Page 12, line 322: suggest. . . elucidation of chemical mechanisms for the formation of processing of aerosol particles. . .

Thank you for this suggestion. We extended the sentence accordingly (l. 331).

New measurement techniques, such as proton-transfer-reaction and chemical ionization mass spectrometry (PTR-MS, CI-MS), have led to the identification of multigeneration products and elucidation of chemical mechanisms  **for the formation and processing of aerosol particles.**

Figure 4 caption: impacts instead of implications?

We clarified and shortened the caption as follows:

Figure 4. Research activities connected to chemical mechanism development for implementation in air quality and climate models.

Page 15, lines 400-409: The authors correctly identify the detection of HO2 and RO2 in chambers as being a valuable addition to the atmospheric chemist's toolbox. However, some more context is probably needed here, because the required instrumentation is complex and expensive, with the result that HO2/RO2 measurements are far from commonplace. In fact, only a handful of facilities can measure HO2/RO2, and while NO3 detection techniques are slightly more prevalent in chambers, instrumentation for measuring absolute concentrations of OH is extremely rare. The development of more sensitive and accessible methods for measuring all types of radicals would be of huge benefit to the field of tropospheric chemical mechanism development.

We agree and have expanded the text to reflect the reviewer's comment. We modified the text as follows and added some references (l. 423):

The huge improvement in the range and sensitivity of chamber instrumentation has led to the ability to comprehensively monitor the chemical evolution of precursor and product species. **This includes the use of CI-MS instrumentation noted above for stable organic species and, although not yet commonplace due to cost and complexity, instrumentation for the measurement of radical intermediates such as $NO_3$, $HO_2$ and $RO_2$ (Fuchs et al., 2008; Onel et al., 2017)**, and OH and $NO_3$ reactivity (Fuchs et al., 2017; Dewald et al., 2020). **Intercomparisons of radical measurements have led to valuable insight of their validity and robustness (e.g. (Schlosser et al., 2009; Fuchs et al., 2010; Dorn et al., 2013).**

Page 16, line 431: Not sure what is meant by sensor networks here. I presume it is referring to a global network of observation sites with advanced instrumentation rather than low-cost sensor networks? Please clarify and rephrase as required.

We added (low cost) in the text to express that the combination of high spatial coverage from satellites with relatively low resolution needs to be accompanied by high(er) resolution measurements that - to date - can be only done by other sensors (l. 452):

Satellites and **(low cost)** sensor networks can provide concentration data with wide spatial (global) coverage.

Page 17, line 450: underlying appears twice in this sentence

Thanks for noting it. We removed it at the first instance (l. 470):

..., since the  assumptions for the underlying algorithms and their evolution are not fully transparent.

Page 18, line 491: Suggest ending the paragraph here and starting a new one for the discussion on the Criegee chemistry.

The referee is right. We merged the two sentences starting with 'Criegee intermediates, formed in the...' with the following paragraph.

Page 18, line 511: Phrasing seems a bit awkward here...how about "Over the last decades, atmospheric chemists have increasingly recognised the need to investigate chemical regimes with more extreme conditions......"

We have rephrased the beginning of section 5.2. as suggested by the referee (l. 535):

Over  recent decades, atmospheric chemists have increasingly recognized the need to investigate  chemical regimes with  more extreme conditions, such as high temperatures in wildfire plumes and low pressures in the free troposphere, but also over large changes in levels and ratios of key pollutants (e.g., $NO_x$, mobile combustion sources).

Page 19, line 527: Refers to the current section in error?

The referee is right. We removed the cross reference to Section 5.2 here.

Page 19, line 528: . . . warmer, dryer climate in some regions of the planet. . .

We made the suggested addition (l. 554):

As global change is evolving resulting in a warmer, dryer climate **in some regions of the planet,** increasing occurrences of wildfires in populated areas represent new challenge

Page 20, line 549: Since this paragraph follows on from anthropogenic VCPs, it is probably better to write ". . . .development also arises from. . . "

We added 'also' as suggested (l. 576):

A second group of new species requiring mechanism development **also** arise

Page 21, line 588: It is more accurate to state that "The mass concentration of PM2.5 is the most commonly used metric to correlate aerosol particles and health effects." In the lines that follow, I think that the role of chemical pathways in affecting the aerosol composition and subsequent health impacts could be made more explicitly.

We changed the first sentence of the paragraph as suggested to make the link between health-related effects and aerosol composition clearer (l. 621). However, we did not add any discussion on health impacts at this point. Also, we did not add further text as a detailed discussion on chemical processes and composition and health impacts would exceed the scope of the paper and journal.

 **The mass concentration of PM2.5 is the most commonly used metric to correlate aerosol particles and health effects.** However, enhanced occurrence of mortality due to cardiorespiratory diseases in the presence of biogenic SOA may point to specific precursors or reaction pathways (Pye et al., 2021) that trigger adverse health effects due to reactive oxygen species (ROS) in aerosol particles (Shiraiwa et al., 2012). Combining such empirical trends in adverse health effects with knowledge on explicit ROS reactions in multiphase systems may be a promising avenue to connect atmospheric chemical processes to (physiological, health-related) target parameters.

Page 22, line 630: Delete "brief"

Agreed. We removed 'brief'.

Page 23, line 655: typo. . . thoroughly

Thanks for catching it. We fixed the spelling.

Page 23, line 676: missing bracket

Thanks for catching it. We added the bracket after $NO_2$.

==========================================
**Referee #2**

The authors provide a nice overview regarding tropospheric chemical mechanisms, their current status and different strategies for mechanism development.

The subject of this opinion paper is highly topical because of the better and better understanding of individual processes based on much better experimental approaches and detection systems over the last while allowing us to run the investigations under close to atmospheric reaction conditions etc. So, e.g., we got a deeper insight into the gas-phase chemistry of RO2 radicals, realized the importance

of RO2 isomerization even for the atmospheric temperature range and its potential importance for generation of SOA precursors, or the importance of RO2 accretion product formation. This changed our view regarding tropospheric degradation processes dramatically in some cases, e.g., for the DMS oxidation characterizing HPMTF as a new main intermediate.

This paper is well structured and easy to read. The authors considered all the different parts needed for successful mechanism development. I think any basic changes are not needed. And it is an "opinion" paper and all authors are well-known experts in their field. Some minor comments for potential improvement of the quality of the paper are already given by other reviewers.

We thank the referee for their very positive and favorable assessment of our paper.
========================================
**Referee #3**

This is a very nicely written Opinion paper that provides a good overview of the state of the field. The title states an aim to describe the current challenges and the resulting needs in the field of tropospheric mechanism development. The paper is structured into six sections, two sections on mechanism development (Strategies and Steps), that function more as a review, a section on recent underpinning research activities, a discussion of Challenges, before concluding and discussing Needs. The challenges are treated in more depth, and the needs are discussed more briefly. I would be happy to see if published as an ACP Opinion, but would suggest that the authors expand the MS somewhat. Firstly to offer more examples of how their community can continue to develop connections with the wider atmospheric community. How can the mechanism community connect better with the needs of field, lab and wider community of composition modellers ? Secondly, to give more input on the outcome of recent work as it relates to best practices for the field, and finally to give more indication of potential for ML/AI techniques in the field.

We thank the referee for their positive and constructive comments. We address the individual comments below.

Section 3 (Steps) provides a good summary of state of field of the chemical mechanism development, and the description of the key area of Structure Activity Relationships is a highlight. The focus of this section (and most of the MS) is on tropospheric gas-phase VOC chemistry, with a less extensive discussion of aerosol chemistry, beyond the liquid phase CAPRAM mechanism for clouds/aerosols. Most the discussion of aerosol chemistry is through its connection to SOA, and this slightly skews discussion away from the implementation of heterogeneous chemistry in CTM/GCM systems where it is necessarily more heavily parameterised, and which should be discussed.

We thank the referee for this suggestion. At multiple places in the text, we point out now the challenges associated with detailed heterogeneous and multiphase chemistry in large scale models:
In the introduction, we added (l. 46):

**Instead, the computational requirements in chemical transport models and global climate models make the use of parameterizations necessary that take into account the phenomenological descriptions of the underlying chemical processes.**

In Section 5.4 (l. 591):

**Given the high computational costs of chemical transport or global models, chemical reactions are often heavily parameterized, e.g., by reactive uptake parameters, that obscure the details of the underlying chemical systems. Such parameterizations do not (necessarily) allow the extrapolation to all conditions covered by the models and, thus, the role and feedbacks of individual chemical processes may be not correctly accounted for. More details and fundamental descriptions of the chemical reactions represents one of the challenges for future model implementations.**

I'd also suggest some extra text added to Section 3.5 and Section 3.6 on mechanism reduction/evaluation for heterogeneous chemistry, and also for aerosol nucleation, as these are key areas where gas phase chemistry connects to aerosol formation and radiative forcing and of intense current interest.

We added the following text to the first paragraph of Section 3.6 (l. 323):

Mechanism development has to occur synergistically with developments of instruments and techniques allowing a more stringent testing of mechanisms for a continuously increasing number of species and processes **that occur in the respective phase (gas, aqueous) or multiphase system.**

The referee is right that we did not specifically mention any gas phase reactions that lead to aerosol nucleation. Thus, we added towards the end of Section 3.6 (l. 336):

**These oxidation products may include autoxidation and/or highly oxygenated organic molecules (HOM) that have been identified to form new particles, e.g. upon $RO_2$ unimolecular reactions, dimer and accretion product formation and/or bimolecular reactions of acylperoxy radicals (Section 3.4).**

We have also added the following at the end of Section 3.4 (see also our response to the community comment by M. Jenkin) (l. 291):

**Furthermore, automated mechanism generators must continue to be updated with new processes as they are discovered and proven to be significant. For example, within the realm of $RO_2$ chemistry alone autoxidation and the formation of organic peroxides (ROOR') and other accretion products in even relatively simple $RO_2/R'O_2$ reactions (Murphy et al., 2023), and bimolecular reactions of acylperoxy radicals (Nozière and Vereecken, 2024) are the subject of current laboratory investigation, but inclusion of these processes leads to unmanageable size of the explicit mechanisms and complicate reduction schemes.**

In fact, a wider discussion on model metrics and model intercomparison would be helpful - metrics are briefly alluded to in section 2 but not returned to. Some idea of the useful metrics for quantifying mechanism's variation between mechanisms in e.g. OPE or across models would be welcome, both between different master mechanisms and the variation between master mechanisms and its reduced forms. This would add value in describing best practice to others in the field, particularly those involved in mechanism reduction for larger scale models. It would be good to hear what the authors consider are the key metrics for such mechanistic activities. It's also important to discuss best practices for identification of critical processes, model sensitivities, and how we might better use the underpinning physicochemical data.

We thank the referee for this suggestion and highlighting the importance of metrics. In general, the choice and usefulness of any metrics depends on the research question to which a chemical mechanism is applied. While OPE is a key metric for air quality models, we do not think that a single (or set of) metric(s) can be given. For example, Barth et al. (2021) compared the concentrations of selected organics in the multiphase model intercomparison. This comparison gave some indication on differences in the chemical multiphase mechanisms while they are of low importance for air quality prediction. Thus, a selection and discussion of such metrics may be very random. To reflect this, we added the following in Section 2 (l. 73):

**The choice of such metrics depends on the specific application of the chemical mechanism and the research questions to be addressed.**
In the second paragraph of Section 5.4 (l. 600):
**Thus, the mechanism comparison metrics in these studies targeted key species for aqueous phase chemistry.**
and at the end of Section 5.4 (l. 621):
**The mass concentration of PM2.5 is the most commonly used metric to correlate aerosol particles and health effects.**

Section 4 on activities related to mechanism development gives recent examples of underpinning activities, and Section 5 is on the Challenges: new insights, new regimes, new compounds, multiphase and FAIR methods. These sections provide a good summary of the limits of the field.

We appreciate the referee's positive assessment.

The MS could be improved with a better description of how the authors feel the community should address the Challenges. I think the manuscript would certainly be improved by more discussion within the text as to how the authors feel the mechanism development community should engage with the wider community, building stronger connections, identifying research questions and exploiting existing work. I suggest that the MS really needs to go further in identifying some specific critical gaps and to make recommendations - how to better exploit work from field, chambers and laboratory studies, how the mechanism development community can add value to these activities.

We thank the referee for this recommendation and appreciate this suggestion. We added text at several places in the manuscript to more strongly illustrate the need for connections between experimental work and chemical mechanisms development:

In Section 5.1 (l. 512): **Mechanism development can also provide data to other fields, e.g. by identifying the best chemical conditions to experimentally probe specific processes (e.g., Kenagy et al. (2024))**

In Section 5.2 (l. 554): **Chemical models incorporating such chemistry can indicate when and where such novel chemistry would affect the atmosphere (e.g. Färber et al. (2024)).**

In Section 6 (l. 730): **In addition, we recommend that chemical mechanism developers should think out of their atmospheric "box" and learn from research efforts being carried out in other related fields. One example is work in the combustion community, where they have combined information from multiple sources (e.g. direct data from lab measurements and theory, indirect data from the field and from different chamber experiments) to produce a mechanism and to assess/target uncertainties (e.g. Olm et al. (2017)).**

Similarly, as they draw on theoretical chemistry work, how can these efforts be made more useful: what level of theory is sufficient for SARs, how to use these studies to reduce explicit mechanisms for CTM/GCM approaches?

We added some indication of the required levels of theory in section 4.1 "Theoretical calculations", mentioning the often-used DFT+CCSD(T) combination, and mentioning the need for post-CCSD(T)/multi-reference calculations for other systems (l. 364). As a discussion of the required level of theory quickly becomes very technical, we consider this a sufficient level of detail for the target audience.

**Many of these reactions are well described by a density functional theory (DFT) characterization of the potential energy system, followed by a higher-level wavefunction refinement of the energies, e.g. by Coupled Cluster, CCSD(T) or equivalent. . . .**

For such systems, **post-CCSD(T) and/or multi-reference calculations are necessary, often at a prohibitive cost, and** the agreement between experiment and theory in terms of barrier heights is currently still poor.

Theoretical studies are generally aimed at explicit mechanisms as they are by their nature highly specific. Theory-based SARs could e.g. be used to lump species based on reactivity or mechanistic similarities rather than molecular similarity. In addition, theory-based SARs can help keep the size of a model smaller by identifying the dominant channels, such that not all possible channels need to be added. Using Criegee intermediates as an example for the latter, we added in Section 5.1 (l. 520):

**Updated theory-based SARs for the chemical reactions of Criegee intermediates may be used to identify the dominant channels and eliminate unnecessary reactions, limiting the size of the mechanism.**

ML forms a part of the backdrop but I would say too much is deferred to future machine learning efforts, and the MS feels rather empty of examples. The MS could discuss what are the factors that make ML/AI approaches suitable. I'd like to see more discussion of work that exemplifies the success of machine learning/AI techniques and I'd reduce the emphasis on the idea that future ML/AI will somehow fix things, as I think the MS would be improved by giving examples of useful datasets, approaches, and tractable problems. In short, where could the available effort most usefully go?

The referee is correct that ML/AI applications in the fields of SARs or theoretical chemistry are currently still a 'promising technique'. The main reason for the rather small progress in this field is the lack of sufficiently complete learning data sets to train algorithms. To express this, we added in Section 4.5 (l. 484):

**However, major developments in their application to SARs or theoretical chemistry are still limited due to the lack of sufficiently complete training data sets.**

================================================

**Community comment by Mike Jenkin**

A chemical mechanism is one of a number of a key components of science and policy models applied to air quality and climate issues - and it is an insurmountable task for an individual modeller to be able to appraise the vast wealth of published information to assemble that mechanism. It is therefore essential that sustainable methods and activities are in place to allow mechanisms to be constructed and freely available to the atmospheric chemistry community, and that such activities are well supported. This opinion paper provides a detailed and comprehensive overview and discussion of the many challenges and requirements for sustainable development of tropospheric chemical mechanisms, authored by a set of experts with considerable expertise and experience in a number of fields relevant to this topic. It thus provides an important benchmark reference and summary document of those methods and activities, which can help to guide and inform future mechanism development strategies.

My previous work in this field largely focused on gas phase mechanism development for organic compounds, mainly through contributions to the Master Chemical Mechanism (MCM). Since that mechanism was first constructed (manually) in the mid 1990s, there have been progressive and considerable developments in understanding of VOC degradation that have required increasing detail to be evaluated and represented in tropospheric chemical mechanisms. I therefore fully concur with the requirements for reliable (and ideally more extensive) evaluated data (sections 3.1 and 4.1); the development of SARs (section 3.2) and generation protocols (section 3.3); and the need to apply automated methods (section 3.4) to allow that detail to be included easily, for mechanisms to be updated regularly and efficiently, and to allow sensitivity tests to be carried out.

The developments in understanding have also highlighted the important requirement for systematic and justifiable methods of mechanism reduction, which are now more important than ever. Accordingly, this is covered in section 3.5, and I agree that this is one of the major challenges for the construction of practical mechanisms that are fully traceable to the fundamental detail derived from laboratory, chamber and theoretical studies. Some particular developments in understanding that have caused a potential explosion of detail in explicit mechanisms, and therefore an increased need for mechanism reduction strategies, are:

- "Autoxidation" mechanisms leading to rapid formation of highly-oxidised organic molecules (HOMs). These mechanisms can involve any number of (1,n) H-shift or ring-closure $RO_2$ isomerisation reactions (sometimes reversible), the rate coefficients for which depend on the precise structure of the species and cannot be represented generically - and which can lead to orders of magnitude increases in mechanism size.

- The formation of large, involatile ROOR' (or other accretion) products from $RO_2$ + R'$O_2$ (permutation) reactions. These cannot be represented easily with the pseudo-unimolecular "peroxy radical pool" parameterisation which is so essential to restrict mechanism size in some mechanisms, including GECKO-A and the MCM.

- New information on the formation and reactions of Criegee intermediates (CIs) formed from $O_3$ + alkene reactions. For example, this includes the very rapid reactions of all stabilised CIs with all organic acids in the mechanism, and the degradation of the hydroperoxyester products (i.e. essentially another class of permutation reaction with large association products).

These mechanistic features are all highlighted in this paper. In view of this, the particular impact of these (and other) features on mechanism size could possibly be given a little more emphasis, along with the resultant related challenges for mechanism reduction.

resultant related challenges for mechanism reduction. We have added some text in Sections 3.4 and 3.6 to further highlight the importance of the Criegee intermediates and the various RO$_2$ + R'O$_2$ reaction pathways:

At the end of Section 3.4 (l. 291): **Furthermore, automated mechanism generators must continue to be updated with new processes as they are discovered and proven to be significant. For example, within the realm of RO$_2$ chemistry alone autoxidation, ROOR' and other accretion products in even relatively simple RO$_2$/R'O$_2$ reaction systems (Murphy et al., 2023), and bimolecular reactions of acylperoxy radicals (Nozière and Vereecken, 2024) are the subject of current laboratory investigation, but inclusion of these processes leads to explosions in size of the explicit mechanisms and complicates reduction schemes.**

In Section 3.6 (l. 336): **These oxidation products may include autoxidation and/or highly oxygenated organic molecules (HOM) that have been identified to form new particles (Bianchi et al., 2019), e.g. upon RO$_2$ unimolecular reactions, dimer and accretion product formation and/or bimolecular reactions of acyl peroxy radicals (Section 3.4).**

**Minor comments**

Page 3, line 73: For consistency with elsewhere, "geckoa" should read "GECKO-A".

Thanks for catching it. We replaced GeckoA by GECKO-A.

Page 7, Figure 3: For consistency with the section title, "Protocols of mechanism generation" should read "Protocols for mechanism generation". There is also an "off" that should be an "of" in the auto generation box.

Thank you for pointing out these mistakes. We fixed the figure accordingly.

[Figure]

Page 19, lines 525-527: I do not fully understand how elevated O$_3$ and suppressed NO$_x$ in megacities might invalidate prevailing understanding of RO$_2$ chemistry. Perhaps this could be explained (note also that the reader is referred to "section 5.2" here, which is the same section). However, I do agree with the general point that changes in the ranges of ambient conditions need to be borne in mind.

We removed the cross reference to Section 5.2 in this sentence. We extended the text to specify the differences in RO$_2$ chemistry as compared to more traditional scenarios (l. 549).

[revised manuscript text omitted]

---

## Editor Decision (ED1)

In general, I suggest making sure the references are listed correctly (I have a few points I noticed below). Also I suggest using "e.g." consistently, perhaps always use "e.g.,"?

- l. 428: remove '(' in front of Schlosser

- l. 514 remove brackets enclosing 2024

- l. 636: write as $C_5H_8$

- l. 657: add citation for CAABA/MECCA ?

- l. 734: no brackets enclosing 2012

- l. 857: change to $HO_2$ (remove 'sub')

- l. 880: give full journal title

- l. 883: give full journal title